# Cystoscopic Guided Laser Cauterization in a Dog with Complete Y-Type Urethral Duplication

**DOI:** 10.3390/vetsci10020126

**Published:** 2023-02-06

**Authors:** Manuel Dall’Aglio, Fausto Quintavalla

**Affiliations:** Dipartimento di Scienze Medico Veterinarie, University of Parma, 43128 Parma, Italy

**Keywords:** laser, urethral duplication, cystoscopic, cauterization, dog

## Abstract

**Simple Summary:**

Duplication of the urethra is a rare congenital malformation in dogs, whose resolution is mainly surgical, but this type of solution is not free from complications. In a case report regarding a young dog, an alternative approach based on diode laser cauterization was tried. Unfortunately, since this method did not provide the desired results, the need for surgery was inevitable.

**Abstract:**

Duplication of the urethra is a rare congenital malformation. A 14-month-old, sexually intact, male, Lagotto dog with complete Y-type urethral duplication was subjected to accessory urethra treatment for cystoscopic guided laser cauterization, with a 10-Watt diode laser with 550-micron fiber and 2.3 Fr outer diameter. The laser cauterization, that was repeated every 14 days for a total of three times, was performed by inserting the instrument from the accessory urethra outlet in the perianal area until it could be seen by the urethroscope, inside the ischial urethra. Nevertheless, this technique that proved to be non-invasive, less expensive than surgery, of short duration, and safe, did not allow the complete closure of the abnormal urethral tract. Therefore, the subsequent surgical removal of the accessory urethra was carried out. To the authors’ knowledge this is the first report on the use of cystoscopic guided diode laser cauterization for accessory urethra treatment.

## 1. Introduction

Duplication of the urethra is a rare congenital malformation observed in humans, dogs, cats, heifers, goats, and llamas [1,2,3], usually diagnosed in young age. Urethral duplication is characterized by the presence of an accessory urethra, either originating from the bladder neck or arising from the primary urethra; it can be complete if the diverticulum has an external opening, or incomplete if the diverticulum is blind ended. Urethral duplication can also be associated with other congenital anomalies.

Excision of the accessory urethra is described as the elective therapeutic option when the orthotopic urethra is patent. In this paper, cystoscopic guided laser cauterization is proposed in a dog with complete Y-type urethral duplication, a technique that has never been utilized before.

## 2. Case Report

A 14-month-old, sexually intact, male, Lagotto breed dog was referred to the Veterinary Teaching Hospital of University of Parma (Italy) with a history of urinary incontinence, with urine coming from the perianal region, as well as from the urethra, during micturition. The dog presented normal urination from the primary urethra (orthotopic), without urine stream alteration, straining or hematuria, pain manifestation or dysuria. The owner reports that the dog maintained a female-like posture during urination until the age of one year, and only since then started lifting the hind limb for urination; the owner also noticed urine leaking occasionally from the perianal region. 

Physical examination of the dog showed normothermia, no lymph node involvement, inspection-palpation of hair coat layers showed no urine-induced burns and genital organs appeared anatomically normal. Complete blood count and biochemistry were within normal limits. Urine had a specific gravity of 1.046, a pH of 6.5, protein level of 23.4 mg/dL, and PU/CU of 0.08. The urine culture yielded was sterile for aerobes, anaerobes, and enrichment. 

After clipping perineal hair, a pinpoint opening was found in the median perineal raphe, approximately 4.5 cm ventral on the left side of the anus (Figure 1). There were no other clinical signs.

An abdominal ultrasound was performed, and no anomalies were found. Under general anesthesia, the penile urethra was catheterized, highlighting that the orthotopic urethra was functional. The retrograde hydropropulsion confirmed the access between the perineal orifice and the orthotopic urethra. To determine the extent of the accessory channel, a contrasted retrograde urethrocystography was performed. The post-administration LLDx projection of 5 mL of radiographic contrast medium iopamidol (Iopamiro 370 mg/mL, Bracco Imaging s.p.a., Milan, Italy), diluted with physiological solution 1:3 allowed for the highlighting of a stripe superimposed on the ischium and in the soft tissues of the perineum, with a length of 6.2 cm and a thickness of approximately 0.32 cm in the craniocaudal direction, originating from the membranous urethra placed between the prostatic portion of the organ and the ischial curvature, and terminating in the perineum, enabling the diagnosis of a complete Y-shaped urethral duplication (Figure 2). 

During the cystoscopic examination performed with a 2.5 mm HugeMed Video-ureteroscope, the penile urethra and ischial curvature appeared normal, with a wide and regular lumen, and free from uroliths and stenotic processes. The mucosa, that appeared regular and normochromic, showed a normal thickness. In the intrapelvic tract of the urethra, between the prostate and the ischial curvature, the outlet of a urethral path was observed (Figure 3). By inserting a catheter from the external perianal orifice, the ostium of the urethra could be easily reached through a length of approximately 5.5 cm. The bladder was normal, with normally formed ureteral ostia.

Under general anesthesia, a cystoscopic guided laser cauterization with 10-Watt diode laser with 550 micron fiber and 2.3 Fr outer diameter (Emmeci 4—Parma, Italy) was performed; the diode laser fiber was inserted from the urethral outlet in the perianal area until it was viewed through the urethroscope, inside the ischial urethra. The cauterization was performed at the starting point of the splitting, using a power of 3 Watts with a continuous pulse, and subsequently proceeding with a spiral movement for the entire length of the abnormal urethral tract, moving back to the outlet on the skin. After 2 weeks, a second treatment was carried out after checking the urethral tract by endoscopic vision. Compared to the previous session, the caliber of the lumen of the path was reduced by half at the starting point of the orthotopic urethra. The therapeutic path was completed with a third intervention 2 weeks later. Upon inspection, the lumen diameter of ostium of the fistula on the urethral wall almost corresponded to the previous treatment, while the cutaneous outlet on the perianal area was almost completely healed, still presenting a flow of urine, although largely decreased.

Given the lack of complete obliteration of the lumen in the abnormal urethral tract, it was decided to proceed with surgery, consisting of legating the origin of the abnormal tract and removing it in its entirety.

The patient was positioned in a sternal decubitus. After a trichotomy of the perianal region and the ventral portion of the abdomen, a purse string suture was performed on the anus. A 2.6-mm urethral catheter was placed in the orthotopic urethra and connected with a closed system to a urine collection bag. Subsequently the perineal ostium of the abnormal urethral tract was catheterized with a 1.3-mm, semi-rigid, urinary catheter. The abnormal urethral tract was delicately isolated, then its resection was performed. The excised portion was sent to the pathological anatomy laboratory for histological examination (Figure 4a–c), which showed that the urethral lumen was lined by transitional epithelium and lamina propria being characterized by a fibro vascular stroma. Two months after surgery, the dog was in good health and urinated normally.

## 3. Discussion

Only seven cases of urethral duplication have been described in dog to date [4,5,6,7,8,9,10]. It was usually found in male dogs and only recently a case has been reported in a female dog [9]. Several anatomical variations exist, depending on the location and opening of the duplicated urethra (perineum or rectum). Urethral duplication may be complete with communication to the skin surface, or may be incomplete and end blindly [8]. Based on anatomical variations, it may show several different clinical manifestations (intermittent stranguria, pollakiuria, haematuria, incontinence, urine dribbling from anus or perineum, and obstructive urolithiasis, sometimes with a concurrent urinary tract infection or secondary dermatitis by urine-induced burns), but in dogs, it can be asymptomatic, without pain manifestation or urination difficulty, as in the present case. According to Stedile et al. [6], the Y duplication is unusual when the orthotopic urethra is normal. Because of the close association between embryonic development of the urogenital and gastrointestinal systems, urethral duplication is almost always accompanied by other duplication anomalies [11,12]. These anomalies result from an abnormal sagittal midline division, and subsequent parallel development of the embryonic hindgut, cloaca, rectum, or urogenital sinus [13]. In the present case, cystourethroscopy confirmed the radiographic findings of a second canal arising from the first, and according to the classification described by Effmann et al. [14], the urethral duplication was of type IIA2, and no other anomalies were highlighted. The differential diagnosis should be made with a urethroperineal fistulae opening adjacent to the anal sphincter, urethrorectal fistula, a rare congenital or acquired condition in dogs with connected lumens of the urethra and the rectum, and recurrent urinary lithiasis [8]. The distinction between urethral fistulas and urethral duplication occurs on the basis of histological findings. The fistula is defined as a structure lined with squamous epithelium, and a truly duplicated urethra is defined as a structure lined with transitional epithelium [6,15], as in the present case. 

Excision of the accessory urethra is described as the elective therapeutic option when the orthotopic urethra is accessible and surgical reconstruction of canine urethral defects involves invasive approaches [7,16,17,18]. Recently Martins et al. [10] reported a major postoperative complication after urethral duplication correction in a dog. In fact, the surgical procedure for urethral duplication is not always necessary. Ultimately, the consensus in human medicine is to pursue treatment for patients with urethral duplication only when they have clinical effects from the condition, such as important functional or cosmetic issues [2].

Interventional urology refers to minimally invasive procedures for the treatment of congenital, acquired, and neoplastic conditions, where endoscopes, fluoroscopy, and other instruments are used to access, under direct visualization, structures inside the urinary tract. Access is most commonly achieved through natural orifices and the procedures are executed internally, with or without minimal external incisions for various diagnostic and therapeutic endeavors [19]. In the last two decades laser techniques have become an increasingly popular method for treatment in human urology, using KTP:YAG (Potassium titanyl phosphate), LBO:YAG (lithium borate), diode lasers, Holmium (Ho):YAG and Thulium (Tm):YAG lasers [20], while in dogs their use is still limited [21]. To the authors’ knowledge, the use of cystoscopic guided laser cauterization in a dog with Y-type urethral duplication, through transurethral technique, had never been reported. Cystourethroscopy is considered a minimally invasive procedure, but does require general anesthesia in order to minimize both patient movement and secondary iatrogenic injury to the lower urinary tract [22]. 

Cauterization is a medical procedure which consists of the therapeutic destruction of tissues using heat or caustic substances. Medical lasers are currently employed in veterinary medicine and surgery, and the laser cautery procedure is a viable option in various clinical situations [23,24]. Laser technology has evolved over time and new concepts and units have emerged. Laser surgery is known to reduce bleeding, inflammation and pain, sterilize the working field, not require the application of sutures and promote healing. The diode laser is a semiconductor device containing gallium, arsenic, and a small amount of aluminum, to produce monochromatic light with a wavelength in the near infrared spectrum. Diode lasers produce cyclodestructive effects by their thermal effects on tissues, causing coagulation necrosis. This occurs with less energy from the diode compared to the Nd:YAG. This results in reduced collateral tissue effects and complications when the diode laser is used [25]. A diode laser is smaller, more efficient and cost-effective in respect to the Nd:YAG laser, can be guided through a flexible quartz fiber and used through an endoscope, and cut and vaporize tissue with a minimal thermal effect [26]. In dogs, the diode laser is mainly used in ophthalmology [25,27], upper respiratory surgery [28,29], and oncology [30,31,32]. Ultrasound-guided endoscopic diode laser ablation holds promise as a palliative treatment for dogs with transitional cell carcinomas (TCC) of the urinary tract, treatment of intramural ureteral ectopia and unilateral ureterovesicular stenosis [33,34,35,36]. Recently, diode laser treatment has been successfully employed in a young dog with congenital tracheoesophageal fistula [37]. In the present case report, the laser technique did not allow the complete closure of the accessory urethra. The failure of cystoscopic-guided laser cauterization could be traced back to the singular progression of ventral and lateral longitudinal musculature and circularly arranged smooth musculature of the urethra in male dogs [38]. The interaction between the laser beam and the tissue depends on physical phenomena, such as reflection, dispersion and absorption [39,40], and this could justify an incorrect penetration in the urethral walls with the laser, the building up of a stenosis, and consequently the inability to correct the abnormal urethral tract.

## 4. Conclusions

Although diode lasers have been available for a long time, their clinical applications are still limited in dogs [41]. The diode laser was used here for the first time to destroy the epithelium and promote scar tissue formation for the closure of an accessory urethra. The technique used in this case proved to be non-invasive, less expensive than surgery, of short duration, and safe, without the complications described when using an open surgical approach. However, in this case report, the laser technique did not allow for complete closure of the accessory urethra. The constant flow of urine in the accessory tract, which persisted between the various treatments, could also have contributed to the lack of therapeutic response, interfering with the cicatricial process. However, there are likely other causes of laser treatment failure that we have not yet identified (for example laser wavelength), and consequently further studies are needed. 

## Figures and Tables

**Figure 1 vetsci-10-00126-f001:**
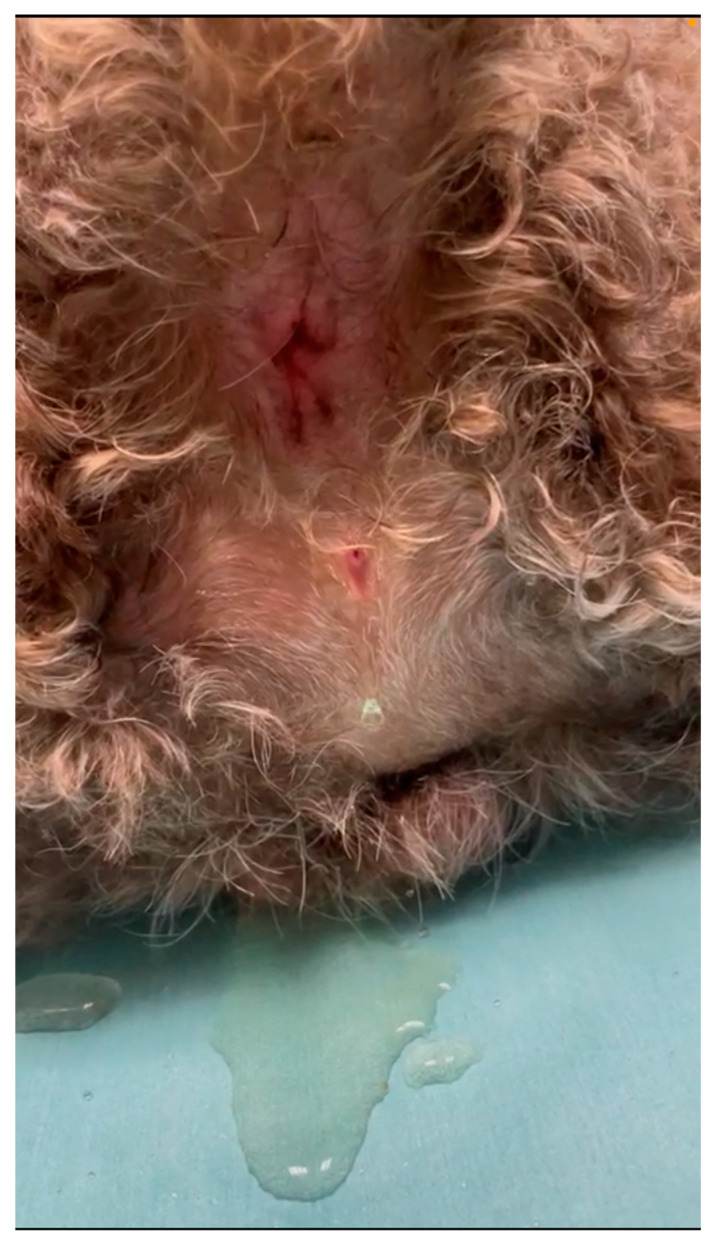
Note the small orifice, from which urine flows, below the anus.

**Figure 2 vetsci-10-00126-f002:**
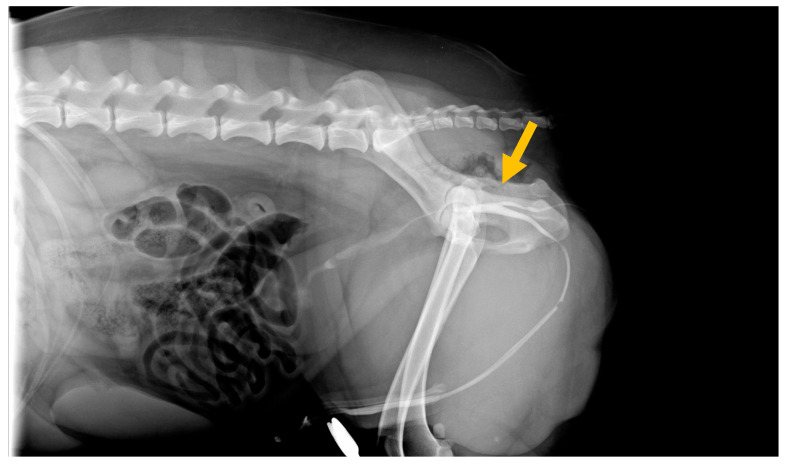
Radiographic image (lateral view) obtained during positive-contrast retrograde urethrocystography, highlighting a normal distal urethra and an accessory urethra. Note the contrast filling of Y-type urethral duplication at time of diagnosis, showing the pelvic urethral opening on the outside.

**Figure 3 vetsci-10-00126-f003:**
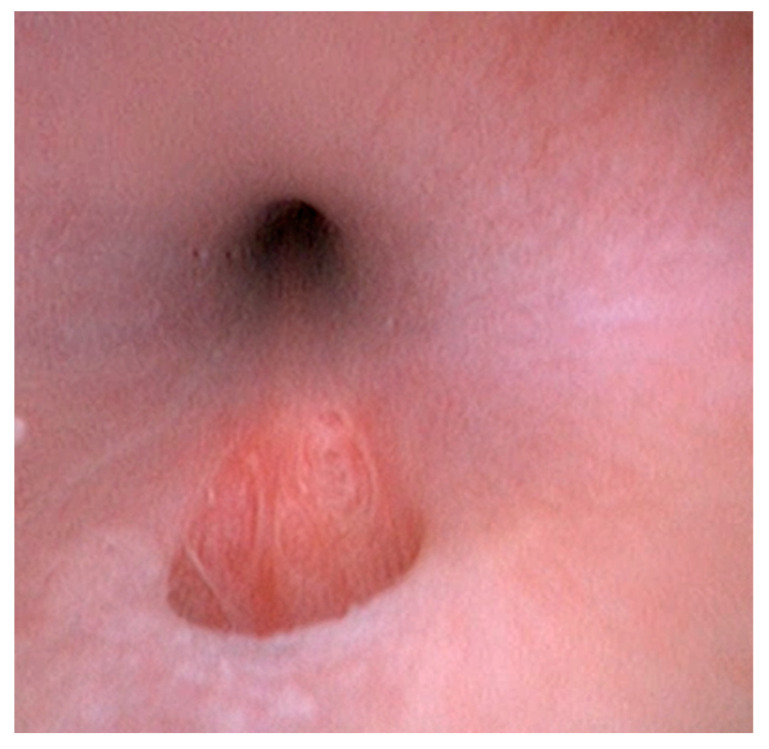
Cystoscopic image. The endoscopic examination revealing a hole (tear-shaped opening on the floor of the urethra).

**Figure 4 vetsci-10-00126-f004:**
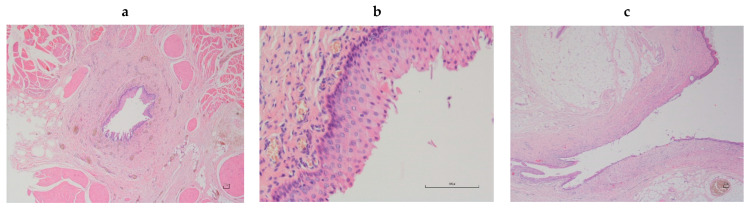
(**a**,**b**)—Istological aspect of a canine accessory urethra: the urethral lumen is lined by transitional epithelium and lamina propria is characterized by a fibro vascular stroma (2X and 20X—E-E); (**c**)—Meatus urinarius of the excised urethra in the present case: the urethra has a transitional epithelial lining that merges with the stratified squamous epithelium at the urethra.

## Data Availability

The data presented in this study are available on request from the corresponding author.

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
