# Peer review of "Cystoscopic Guided Laser Cauterization in a Dog with Complete Y-Type Urethral Duplication"

_vetsci, 2023, doi:10.3390/vetsci10020126_

Round 1

Reviewer 1 Report

Comments to the authors

I recommend a careful proofreading of the text for the typing errors and by a English native speaker to improve the quality of the manuscript.

The Table 1 doesn’t bring any additional interest that the sentence within de manuscript. I would remove this table.

Figure 2: could you modify the position of the arrow to precisely show the duplicated urethra?

Line 111: trichotomy: Do you want to say hair clipping?

Line 112: tobacco bag suture: Do you want to say purse string suture?

Line 113: anal sphincter: anus, I guess.

Line 119: urethral lume: urethral lumen, I guess.

Line 175: Ndeye laser: Nd:YAG, I guess

Do you think that the use of fluoroscopy could have improved your technique? Please, develop within the Discussion.

Author Response

I recommend a careful proofreading of the text for the typing errors and by a English native speaker to improve the quality of the manuscript.

The Table 1 doesn’t bring any additional interest that the sentence within de manuscript. I would remove this table.  Done 

Figure 2: could you modify the position of the arrow to precisely show the duplicated urethra? Done

Line 111: trichotomy: Do you want to say hair clipping? Yes, understood as a wide shave of the coat to visualize the accessory meatus
Line 112: tobacco bag suture: Do you want to say purse string suture? Yes

Line 113: anal sphincter: anus, I guess. Thanks for suggesting this error.

Line 119: urethral lume: urethral lumen, I guess. Thanks for suggesting this error. 

Line 175: Ndeye laser: Nd:YAG, I guess Thanks for suggesting this error.

Do you think that the use of fluoroscopy could have improved your technique? Please, develop within the Discussion.

Fluoroscopy is a particularly useful diagnostic imaging method in our opinion and that certainly could be used, but sometimes fluoroscopy images are subject to spatial blurring and also temporal blurring. Being in possession of a video-ureteroscope capable of providing us with excellent quality images, we were able to view the entire length of the urethra and visualize the origin of the anomalous urethral tract in the best possible way, furthermore, it was possible to evaluate on the urethral mucosa, the degree of scarring and changes that occurred after each laser treatment, which fluoroscopy would not have allowed to do.

Reviewer 2 Report

The manuscript clearly presents this clinical case of a rare condition, yet very interesting. The combined use of first-level diagnostic, i.e., the urethra-cystography and the "fistulography" with the cystoscopy, is fascinating. Anyhow, the low efficacy of the laser treatment is the major pitfall of the manuscript, even after three applications of the technique. Since no reports, to the best of my knowledge, have described the use of the laser to correct this or similar malformations, I think the Authors should deepen more in the discussion about the possible causes that led to the failure of the technique - as they said, power of the laser? Would a fourth application have led to the complete resolution? Any preclinical study about the efficacy of the laser in similar models? Finally, I would suggest English language editing.

Author Response

The manuscript clearly presents this clinical case of a rare condition, yet very interesting. The combined use of first-level diagnostic, i.e., the urethra-cystography and the "fistulography" with the cystoscopy, is fascinating. Anyhow, the low efficacy of the laser treatment is the major pitfall of the manuscript, even after three applications of the technique. Since no reports, to the best of my knowledge, have described the use of the laser to correct this or similar malformations, I think the Authors should deepen more in the discussion about the possible causes that led to the failure of the technique - as they said, power of the laser? Anatomic anomalies (eg, ectopic ureters, vestibulovaginal septal remnant, urethral septum) are corrected using a Holmium:YAG or diode laser (Marie Llido, Catherine Vachon, Melanie Dickinson, Guy Beauchamp, Marilyn Dunn (2020):Transurethral cystoscopy in dogs with recurrent urinary tract infections: Retrospective study (2011-2018). J. Vet Internal Med.  34(2):790-796). Since there are no experiences like ours in the bibliography, we can only make hypotheses as reported in the conclusions. It is possible that the failure of our technique could be traced back to the individual course in male dogs of the ventral and lateral longitudinal musculature and the circularly arranged smooth musculature of the urethra. The constant passage of urine in the accessory tract which persisted between the various treatments could also have contributed to the lack of therapeutic response, interfering with the cicatricial process.

Would a fourth application have led to the complete resolution? Apart from an initial therapeutic response, no change was observed in the two subsequent treatments, therefore it is conceivable that even a fourth treatment would have been useless. In order to avoid an obstinate treatment and in mutual agreement with the owner, we then proceeded with the surgical treatment as proposed up to now by all the authors who have dealt with the problem of urethral duplication.

Any preclinical study about the efficacy of the laser in similar models? To our knowledge, there are no published experiences in both the human and veterinary fields in similar models, except for other anatomical structures of the urinary tract.

Finally, I would suggest English language editing. 

Round 2

Reviewer 2 Report

Lines 50-58 are a little confusing: the authors may try to order all CBC, biochemical and urine results and then physical examination.

English has been improved, but still poor.

Author Response

Lines 50-58 are a little confusing: the authors may try to order all CBC, biochemical and urine results and then physical examination.

Response 1: Thank you for reporting this inaccuracy. We have arranged the sentence according to your valuable information

English has been improved, but still poor.

Response 2: Thanks for your advice. We have revised the English text and submitted it to an English native speaker with the hope of having improved the quality of the manuscript. The most recent corrections made are highlighted in yellow